# Rhizosphere Microbiome Co-Occurrence Network Analysis across a Tomato Domestication Gradient

**DOI:** 10.3390/microorganisms12091756

**Published:** 2024-08-24

**Authors:** Mary M. Dixon, Antisar Afkairin, Daniel K. Manter, Jorge Vivanco

**Affiliations:** 1Department of Horticulture and Landscape Architecture, Colorado State University, Fort Collins, CO 80523, USA; mary.dixon@colostate.edu (M.M.D.); antisar.afkairin@colostate.edu (A.A.); 2United States Department of Agriculture–Agricultural Research Service, Soil Management and Sugar Beet Research, Fort Collins, CO 80526, USA; daniel.manter@usda.gov

**Keywords:** tomato, phosphorus, co-occurrence network, crop domestication

## Abstract

When plant-available phosphorus (P) is lost from a soil solution, it often accumulates in the soil as a pool of unavailable legacy P. To acquire legacy P, plants employ recovery strategies, such as forming associations with soil microbes. However, the degree to which plants rely on microbial associations for this purpose varies with crop domestication and subsequent breeding. Here, by generating microbial co-occurrence networks, we sought to explore rhizosphere bacterial interactions in low-P conditions and how they change with tomato domestication and breeding. We grew wild tomato, traditional tomato (developed circa 1900), and modern tomato (developed circa 2020) in high-P and low-P soil throughout their vegetative developmental stage. Co-occurrence network analysis revealed that as the tomatoes progressed along the stages of domestication, the rhizosphere microbiome increased in complexity in a P deficit. However, with the addition of P fertilizer, the wild tomato group became more complex, surpassing the complexity of traditional and modern tomato, suggesting a high degree of responsiveness in the rhizosphere microbiome to P fertilizer by wild tomato relatives. By illustrating these changing patterns of network complexity in the tomato rhizosphere microbiome, we can further understand how plant domestication and breeding have shaped plant–microbe interactions.

## 1. Introduction

Phosphorous fertilizer is mined from nonrenewable rock phosphate which may deplete in reserves within a century [1]. Thus, it is important to elucidate mechanisms of soil P transfer (from unavailable to available forms) to reduce the application of fertilizer sourced from rock phosphate. Beneficial soil microbes, for example, can increase soil P bioavailability by direct solubilization of precipitated phosphates and exuding extracellular enzymes such as phosphatases [2,3]. An important extracellular enzyme is alkaline phosphatase, which hydrolyzes organophosphate esters and makes recalcitrant forms of organic soil P plant-available [2]. Bacteria that produce alkaline phosphatase improve P absorption by plant roots [4], and genes that encode for alkaline phosphatase are widespread across prokaryotic soil phyla [2]. Some soil bacteria also contain a gene-expressing pyrroloquinoline quinone synthase (*pqqC*), and these bacteria are capable of soil phosphate solubilization [5]. The *pqqC* gene plays a role in the synthesis of the cofactor pyrroloquinoline quinone [6], which has several beneficial functions including plant growth promotion, systemic plant defense induction, and chelation of complexed phosphates [5]. 

A strategy for plant roots to acquire sparingly available soil P is to form associations with beneficial soil bacteria, such as those capable of phosphate decomposition and solubilization [7]. Roots form these associations by releasing chemoattractant compounds to select for certain beneficial microbes [8]. Attracting these bacteria alters the rhizosphere microbiome, a dynamic environment with diverse and populous microbial communities that is essential to plant health [8,9]. However, microbial associations with plant roots are highly influenced by the host plant as different plant species and cultivars can associate with distinct microbes [9,10,11,12]. For example, while the composition of bacterial endophytes did not change between isogenic lines (expression and non-expression of *cry* genes), there were clear differences in root-associated microbial communities among different maize varieties [13]. Genotypic variability in this capability has been observed in many crops including cotton [14], maize [15], wheat [16], melon [17], and lima bean [18].

Beyond inter- and intra-specific variation in microbial associations, there is also noted divergence in rhizosphere microbiome compositions as a result of domestication and plant breeding events [19]. For instance, heirloom varieties of tomato harbor greater total microbial biomass in the rhizosphere when compared to modern accessions [20]. Further, the rhizosphere bacterial community of domesticated tomato varies from that of a wild progenitor, and these differences in the rhizosphere have been shown to amplify across successive planting of either domesticated or wild tomato [21]. Notably, wild tomato has also been shown to exude acid phosphatase to a greater degree [22] and more readily associate with P-solubilizing bacteria [12] than domesticated tomato. Similar patterns of divergence have been noted in other crops. Among maize accessions, it has been shown that landrace and inbred lines have greater alpha-diversity than wild teosinte [23]. These differences in the rhizosphere microbiome of wild and domesticated crops may relate to a disruption in the reliance on microbes to acquire nutrients from the soil. Domesticated tomato plants, for example, are not only less responsive to beneficial soil–bacterial associations [24], but they are also more likely to have negative soil–plant feedback than wild tomato [25]. 

Moreover, wild tomato relatives have been shown to be resilient to low-P stress [22,26]. In comparing the root-associated communities of a representative wild and domesticated tomato in P-depleted conditions, it was shown that not only were there differences in alpha- and beta-diversity, but there were clear functional differences between these domestication groups. Wild tomato is more intimately associated with P-solubilizing bacteria than domesticated tomato, particularly in the endosphere [26]. Similarly, in conditions of low-P stress, wild tomato [26] increased the relative abundance of P-decomposing bacteria compared to domesticated counterparts. These findings are valuable because low-P stress is a major driver of changes in the rhizosphere bacterial communities [27,28], and these microbes affect plant growth in conditions of nutrient stress [29,30]. Beneficial plant growth-promoting bacteria, including strains of *Bacillus* and *Arthrobacter*, have been shown to solubilize P in vitro, and inoculations increase tomato biomass in conditions of P deficiency and high salt stress [31]. However, the interactions occurring among rhizosphere bacteria associated with low-P-stressed crops are poorly understood. A method to elucidate microbial co-oscillations and associations affected by environmental disturbance is through the generation of co-occurrence networks [32]. Thus, we aimed to utilize co-occurrence networks to explore the interactions occurring among bacterial groups in the rhizosphere of low-P-stressed wild, traditional, and modern tomato. We hypothesize that network connectivity and microbial assemblages have changed across a tomato domestication gradient and are more affected by domestication than by soil nutrition.

## 2. Materials and Methods

The methods describing plant and soil selection, tomato growth conditions, DNA extraction, and PCR and amplicon sequencing presented in this manuscript are replicated from Dixon et al. [12]. Thus, all raw data used in this manuscript are culled from Dixon et al. [12]. However, all analyses and datasets presented here are new. A brief description of the replicated methodology is delineated below as well as a comprehensive description of novel in silico analysis. 

### 2.1. Plant and Soil Selection

Four accessions were selected to represent each of the following tomato domestication groups: wild [*Solanum pennellii* (LA0716), *S. pimpinellifolium* (LA1580), and *S. lycopersicum* var. *cerasiforme* (LA1519, LA1698)], traditional (‘Rutgers’ and ‘Brandywine Pink’, ‘Matchless’ and ‘Marglobe’), and modern (‘Bobcat’, ‘Quali T’, Syngenta experimental line A, and Syngenta experimental line B). Seeds from these selected varieties were retrieved from the University of California Davis Tomato Genetic Resources Center (TGRC) (Davis, CA, USA) (all wild accessions), the W. Atlee Burpee Co. (Warminster Township, PA, USA) (‘Rutgers’ and ‘Brandywine Pink’), the Victory Seed Co. (Irving, TX, USA) (‘Matchless’ and ‘Marglobe’), or through a donation from the Syngenta Co. (Basel, Switzerland) (all modern accessions). 

Field soil was collected from the Agricultural Research, Development, and Education Center in Fort Collins, CO, USA. Soil was collected from the top 20 cm, avoiding topsoil. The collected soil was mixed at a 1:1 volume-to-volume ratio with sand (Quikrete Play Sand, Atlanta, GA, USA) to reach a final bioavailable P concentration, as approximated by the Olsen P and bicarbonate extraction [33], of 2.6 mg/kg. 

### 2.2. Tomato Growth Conditions

To improve germination rates and decrease the time to germination, seeds were surface-sterilized following the recommendation of the TGRC [34]. Surface-sterilized seeds were placed in Petri dishes on filter paper moistened with DI water. Once the seeds germinated, they were transferred to a low-nutrient potting mixture (PromixBK, Québec, QC, Canada). Seeds were kept in this potting mixture until the cotyledons fully expanded (approximately two weeks). At this point, the seedlings were transferred to pots, which contained the sand–soil mixture. Experimental units were grown in a greenhouse at Colorado State University which had the following environmental conditions: 18–21 °C daily temperature range, 17–20 °C nightly temperature range, and 16 h photoperiod. The pots (dimensions: 11 cm width, 11 cm length) were arranged in a completely randomized design as determined by an open-source randomizing software [35], and each treatment had ten replicates (*N* = 240). The fertilization treatment was applied one week after transplantation. Half of the experimental units received sufficient P fertilization levels through the application of triple superphosphate (TSP) (46% P_2_O_5_) at rates equivalent to 163 kg/ha P_2_O_5_. To not conflate P deficiency symptoms with N deficiency, we fertilized all experimental units to reach N sufficiency using polymer-coated urea (44% N) at rates equivalent to 118 kg/ha N. Plant growth was terminated eight weeks after fertilization. 

### 2.3. Collection and DNA Extraction of Rhizosphere Soil Samples

Plants and attached soil were removed from pots and were gently shaken. Following the methodology of Pantigoso et al. [36], soil that remained adhered to the roots was defined as rhizosphere soil. Rhizosphere soil from each replicate (*N* = 240) was collected in sterile 15 mL centrifuge tubes and stored at −80 °C until DNA extraction. Following the manufacturer’s instructions, gDNA was extracted using the DNeasy Power Soil Pro kit (Qiagen, Hilden, Germany) and the automated QIAcube (Qiagen, Hilden, Germany). 

### 2.4. PCR and minION Sequencing Conditions

All DNA extracts from all soil samples underwent PCR and minION sequencing. DNA samples were fluorometrically quantified using a Qubit (Thermo Scientific, Rockford, IL, USA) to determine the dilution factor of DNA extracts with sterile nuclease-free water (1-to-20 dilution). Each well contained 4 µL diluted DNA extract, 20 µL Phusion HSII master mix (Thermo Scientific, IL, USA), 14.4 µL nuclease-free water, and 0.8 µL of each forward (27F Bacterial Mn, 5′-TTTCTGTTGGTGCTGATATTGC AGRGTTYGATYMTGGCTCAG-3′) and reverse (1492 Universal Mn, 5′-ACTTGCCTGTCGCTCTATCTTC TACCTTGTTACGACTT-3′) primer. The entirety of the V1-V9 region of the bacterial 16S rRNA gene was amplified. This mixture was prepared in QIAgility (Thermo Scientific, IL, USA). Amplification reactions were held in a Roche LightCycler 96 (Roche Sequencing Solutions, Indianapolis, IN, USA) for 30 s at 98 °C followed by 25 cycles of 15 s at 98 °C, 15 s at 50 °C, and 60 s at 72 °C. There was a final extension for 5 min at 72 °C. Paramagnetic beads (AMPure XP beads, Beckman Coulter, Brea, CA, USA) with 70% EtOH rises were used to purify the PCR products. The PCR products were diluted with sterile, nuclease-free water to reach final DNA concentrations of 4 ng/µL. Using the PCR Barcoding Expansion kit (Oxford Nanopore, Oxford, UK), 1 µL of sample-specific barcodes was added to 5 µL of each purified, diluted PCR product in a new PCR plate. In each well, 25 µL Phusion HSII master mix and 19 µL sterile, nuclease-free water were added. Amplification reactions for this plate included an initial cycle at 98 °C for 30 s followed by 15 cycles of 98 °C for 15, 62 °C for 15 s, and 72 °C for 60 s. The final extension for this reaction was for 5 min at 72 °C. 

To prepare the library for sequencing, PCR products were pooled together and purified using paramagnetic beads and 70% EtOH. The Ligation Sequencing Kit V14 (SQK-LSK114) (Oxford Nanopore Technologies, Oxford, UK) was used, following the manufacturer’s instructions, to ligate samples. The ligated, pooled samples were fluorometrically quantified and diluted to reach 20 ng/µL DNA. The minION flow cell (Oxford Nanopore Technologies, Oxford, UK) was primed with a flash buffer, and 50 mM of the library was added. Signal data were collected with the MinKNOW (Oxford Nanopore Technologies, Oxford, UK, website: https://community.nanoporetech.com/docs/prepare/library_prep_protocols/experiment-companion-minknow/v/mke_1013_v1_revdc_11apr2016 accessed on 1 March 2023) software across 48 h. Guppy (Oxford Nanopore Technologies, Oxford, UK) was used to base call and demultiplex the signal data and filter to a 70 q score. Emu [37] was used to generate taxonomic profiles. Data were filtered to remove samples with fewer than 10,000 reads.

### 2.5. Co-Occurrence Network Generation

Data were filtered to remove any samples with less than 10,000 reads. Networks were developed in RStudio (version 4.1.2) using the igraph (version: 2.0.3) and microeco (version: 1.8.0) [38] packages. Four networks were generated: one representing all samples in unfertilized soil and three representing each tomato domestication group in unfertilized soil. Spearman’s correlation was calculated using a filter threshold of 0.001, and a correlation optimization was used to identify the correlation threshold. Networks were constructed with a *p*-value threshold of 0.01 with an FDR *p*-value adjustment. Network modules were identified using the “cluster fast greedy” method, which employs a hierarchical agglomeration algorithm to quickly detect the structures of communities within the network [39]. Generated networks were exported from RStudio using the rgexf package (version 0.16.3) [40]. These co-occurrence networks were visualized using Gephi (version 0.10.1) [41]. The Fruchterman–Reingold force-directed algorithm—which draws undirected graphs with straight edges by use of extant repulsive forces among adjacent and non-adjacent nodes [42,43]—was used for network layout. Nodes and edges were colored by the bacteria phylum. The size of nodes was proportionate to the degree number. The chorddiag package (version 0.1.3) [38] was used to construct chord diagrams for the top ten most influential phyla.

### 2.6. Data Analysis

Keystone taxa were identified following Qiu et al. [44]; briefly, keystone taxa were those that had degree values > 6, weighted degrees > 6, closeness centralities > 0.14, clustering coefficients > 0.09, and betweenness centralities < 0.05. To determine predictive functions that were present in the keystone taxa, PICRUSt2 [45] was used to identify KEGG orthologs that mapped the bacterial species to genes of interest. Bacteria were classified into the following functional categories as determined by the presence of the corresponding enzymes and enzyme commission (EC) numbers listed parenthetically: stress (indolepyruvate decarboxylase, EC 4.1.1.74), siderophore (isochorismate synthase, EC 5.4.4.2), P decomposition (alkaline phosphatase, EC 3.1.3.1), N decomposition (leucyl aminopeptidase, EC 3.4.11.1; urease, EC 3.5.1.5; amidase, EC 3.5.1.4), dissimilatory nitrate reduction (nitrite reductase, EC 1.7.2.2), C/N cycling (N-acetyl-B-glycosaminidase), C decomposition (alpha-glucosidase, EC 3.2.1.20; beta-glucosidase, EC 3.2.1.21), biocontrol (S,S-butanediol dehydrogenase, EC 1.1.1.76), and assimilatory nitrate reduction (ferredoxin–nitrite reduction, EC 1.7.7.1). A two-way ANOVA was used to determine differences in the relative abundance of keystone taxa among the three domestication groups. A Tukey’s HSD test was used for pairwise comparisons. Separate ANOVA tests were run for the fertilized group and the unfertilized group to determine differences in the presence of keystone taxa for those individual fertilization groups. 

## 3. Results

### 3.1. Soil Bacterial Community Interactions and Functionality

In unfertilized soil, bacterial communities demonstrated many strong and positive co-occurrence relationships in the tomato rhizosphere (Figure 1). In particular, the Pseudomonadota phylum was positively correlated with the populations of many bacterial groups, including Actinomycetota, Bacteroidota, Bacillota, Planctomycetota, Acidobacteriota, Chloroflexota, Gemmatimonadota, Candidatus Saccharibacteria, Nitrospirota, Verrucomicrobiota, and other Pseudomonadota members (Figure 1C,D). Pseudomonadota was also the most influential phylum as illustrated in the co-occurrence network, comprising 21.6% of the connections in the unfertilized network (Figure 1A) and 25.6% in the fertilized network (Figure 1B). Following Pseudomonadota in unfertilized soil, Actinomycetota, Bacillota, and Bacteroidota were the most influential phyla, consisting of 20.4%, 12.0%, and 9.6% of the network connections, respectively. Similarly, in fertilized soil, the most influential phyla after Pseudomonadota were Actinomycetota, Bacteroidota, and Bacillota with 25.6%, 11%, and 9.8% of the network connections. Only significant positive correlations were identified. The unfertilized network consisted of 167 nodes and 2004 edges with an average weighted degree of 18.6. The fertilized network had similar measurements: 164 nodes, 1244 edges, and an average weighted degree of 12.4. The co-occurrence network for the unfertilized soil showed a strong modular structure, as indicated by a clustering coefficient of 0.728 and modularity of 0.285. For the network in the fertilized soil, there was a clustering coefficient of 0.720 and a modularity of 0.420.

From these networks, 110 and 84 bacteria were identified as highly influential keystone taxa in unfertilized and fertilized networks, respectively (Appendix A). These keystone taxa were identified to have diverse predicted functions, including those involved in nutrient cycling, biocontrol, and stress (Figure 2). In both unfertilized soil (Figure 2A) and P-fertilized soil (Figure 2B), there were no significant changes in the relative abundance of keystone taxa with different predicted functions, thus suggesting high diversity in the functions of keystone taxa. The identified keystone taxa showcased low relative abundance, with median compositions of 0.4% in the rhizosphere of both unfertilized soil and fertilized soil (Figure 2). 

### 3.2. Tomato Domestication Influence on Soil Bacterial Interrelationships

The abundance of the identified keystone taxa in fertilized and unfertilized soil varied with tomato domestication (Figure 3). Modern tomato accumulated a greater relative abundance of the keystone taxa compared to wild tomato in unfertilized soil (Figure 2A). In unfertilized soil, the relative abundance of keystone taxa did not differ between traditional tomato and the other domestication groups (Figure 3A). However, in fertilized soil, there were no differences in the relative abundance of keystone taxa (Figure 2B). 

To further elucidate changes occurring in the rhizosphere microbiome of wild, traditional, and modern tomato, we ran individual co-occurrence networks for each domestication group. Here, we observed a varied structure of networks within each tomato domestication group (Figure 4). In unfertilized soil, the wild tomato rhizosphere (165 nodes, 1071 edges) had a less complex network than the other domestication groups (Figure 4A), with smaller node and edge numbers relative to the other traditional (174 nodes, 1708 edges) and modern (181 nodes, 2198 edges) groups. Traditional tomato had a relatively moderate network complexity (Figure 4B), and modern tomato had the highest network complexity (Figure 4C) in unfertilized soil. However, in fertilized soil, the wild tomato group increased node and edge number and became the most complex network (193 nodes, 1613 edges) (Figure 4D). In fertilized soil, traditional tomato had the least complex network (175 nodes, 1173 edges) (Figure 4E), and modern tomato had moderate complexity (182 nodes, 1565 edges) (Figure 4F). However, the most influential phyla were consistent across tomato domestication groups and fertilization, with Pseudomonadota correlating to the greatest number of other phyla in the rhizosphere (Figure 4). 

## 4. Discussion

The co-occurrence network analysis of tomato domestication groups—wild, traditional, and modern—in low- and high-phosphorus (P) conditions revealed insightful information into varied rhizosphere microbial interactions. Here, we found that the network complexity and structure of the microbiome in unfertilized and fertilized soil was similar. Further, the most highly connected phyla were the same in both unfertilized and P-fertilized soil. However, in examining the network structure of the different tomato domestication groups, there were clear differences. We found that with the application of P fertilizer, the wild tomato domestication group increased network complexity.

While other research has not been conducted on rhizosphere co-occurrence network complexity across tomato domestication, there is a similar study conducted on rice with which our results are consistent. The finding that there is higher network complexity in wild tomato compared to lower network complexity in traditional tomato is in congruence with Chang et al. [46] who found that the microbial network complexity of wild rice was greater than its domesticated counterpart. Further, Sun et al. [47] found that wild accessions of rice have a greater abundance of bacterial chemotaxis genes than modern rice. This difference in chemotaxis between the two domesticated groups of rice is a result of changing metabolite profiles of root secretions that were affected by plant domestication [47]. Domestication was shown to reduce the abundance of sugars and alcohols in rice root exudates [47], and sugar has been shown to be a major chemoattractant for beneficial soil bacteria [48]. Sugars and other high organic C compounds that are exuded by plant roots strengthen interactions in the rhizosphere [49], and microbial interactions are approximated by measurements of co-occurrence network complexity [50]. Thus, the changing root exudate profiles may have led to greater network complexity in wild compared to domesticated crops.

While the rhizosphere network complexity was heightened for wild relatives compared to traditional and modern tomato in P-fertilized soil, it was diminished in unfertilized soil. Conversely, modern and traditional tomato showed little to no change in their network characteristics between the unfertilized and fertilized treatments. This strong response of the rhizosphere microbiome of wild tomato to P fertilizer compared to modern and traditional tomato may be explained by wild tomato being more responsive to P fertilization. Wild tomato has been shown to increase biomass production in response to P fertilizer to a greater degree than domesticated tomato [12]. In potato, a crop closely related to tomato, it was also shown that non-cultivated accessions were highly responsive to the addition of P fertilization compared to cultivated varieties [36]. Thus, wild tomato and its associated microbiota may be more sensitive to varied levels of exogenous P, and, therefore, more subject to change with P additions than domesticated varieties.

In addition to wild tomato varying from its domesticated counterparts, modern tomato and traditional tomato also had marked differences in network complexity; the modern tomato microbiome was more complex than traditional tomato. These differences could be explained by tomato breeding patterns over time. In tomato breeding programs, there has been a heightened emphasis on the introgression of traits from wild relatives since the 1970s [51], and our tested accessions of traditional tomato were developed prior to that time. It is therefore possible that the increased genetic diversity of modern tomato relative to heirloom tomato resulted in an increased complex co-occurrence network of modern tomato accessions when compared to earlier cultivars. While genetic erosion may have occurred when tomatoes were domesticated and developed for human use [52], continual introgressions of wild genetic material into modern germplasm has increased the genetic diversity of current tomato cultivars [51,53,54].

The altered complexity we observed in our tested accessions may also indicate variations in microbial diversity. Network complexity and microbial diversity have been shown to be related in a study examining tomato root-associated environments [55]. The authors found that in compartments with low alpha-diversity, there was a concomitantly low measure of network complexity [55]. There has been much research into the changing bacterial diversity patterns as a function of crop domestication [19,56,57,58]. Although recent research suggests that alpha-diversity measurements do not differ among wild, traditional, and modern tomato [12,59], that pattern is not true for all crops. For example, while domesticated wheat harbors greater bacterial diversity, its fungal diversity is less compared to wild wheat [60]. Therefore, network complexity and diversity may be affected by domestication and major plant breeding events. For example, breeding events, such as the Green Revolution, have taken place since the development of our tested traditional tomato accessions. The Green Revolution involved the development of semi-dwarf, high-yielding crops concomitant with increased applications of fertilizers and pesticides [61]. These altered agricultural practices may have significantly influenced the diversity and composition of these microbial communities [61,62], which suggests the capability of plants to enrich microbes with functions that are advantageous under specific management regimes [63]. Such findings emphasize the influence of plant breeding and selection on microbial dynamics within the rhizosphere. The observation that modern crops maintain microbial network complexities comparable to wild species challenges previous notions. This realization fosters a deeper understanding of how selective trait selection by breeders can positively affect crop–microbe interactions, promoting improved resilience, plant health, and nutrient cycling of soil ecosystems [60].

Soil nutrient cycling is highly influenced by the presence of influential, keystone taxa [64]. In the current study, Pseudomonadota was associated with the greatest change in other bacteria in the tomato rhizosphere. This finding suggests that beneficial bacteria may be highly influential in rhizosphere microbial assemblage because many plant-growth-promoting bacteria are present in Pseudomonadota, and the genetics and beneficial functions of this phyla have been extensively studied [65,66,67]. Further, we identified numerous keystone taxa with diverse predicted functions to be present in both fertilized and unfertilized soil, thus indicating that the bacteria that are most influential in the rhizosphere microbiome are also capable of facilitating many different soil processes. Although these keystone taxa are highly influential, they comprise a small proportion of the rhizosphere microbiome. This finding is in agreement with Shi et al. [68] who found that keystone microbes tended to have low abundances and concluded that taxa with low abundances can highly influence the functionality of the rhizosphere.

## Figures and Tables

**Figure 1 microorganisms-12-01756-f001:**
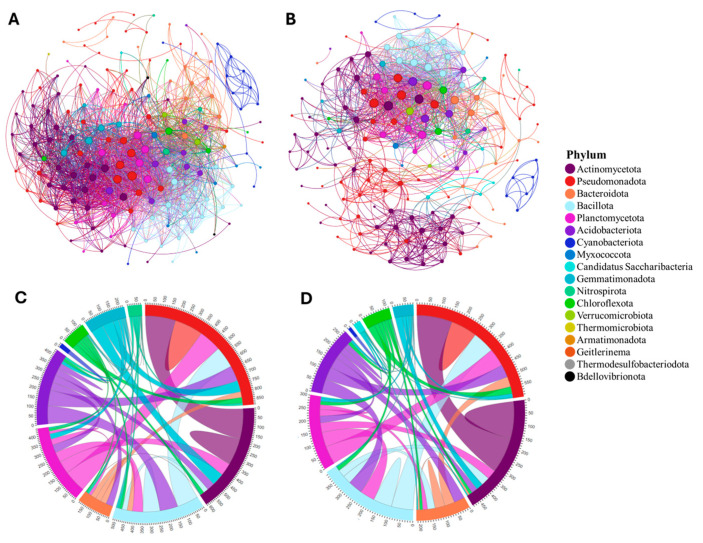
Interrelationships of bacteria in unfertilized soil of the tomato rhizosphere as illustrated through chord and co-occurrence network diagrams. (**A**) Co-occurrence network diagram in unfertilized soil. (**B**). Co-occurrence network diagram in fertilized soil. (**C**) Chord diagram in unfertilized soil. (**D**) Chord diagram in fertilized soil. Color denotes bacterial phylum. Node size is proportional to degree (i.e., number of significant correlations). Only bacteria that were significantly correlated (optimized Spearman’s correlation > 0.72, *p* < 0.01) are included.

**Figure 2 microorganisms-12-01756-f002:**
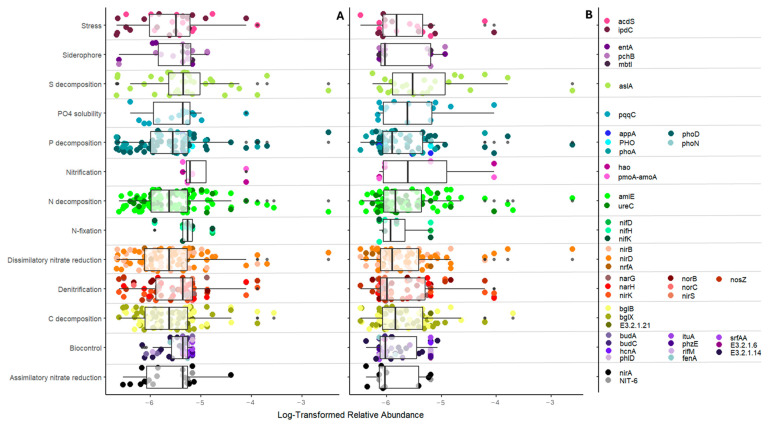
Log-transformed relative abundance with mapped predicted functions of keystone taxa. Predicted functions were identified using PICRUSt2. Each dot represents a taxon with the corresponding gene on the legend in the right panel and the predictive function on the *y* axis. (**A**) Keystone taxa relative abundance in unfertilized soil. (**B**) Keystone taxa relative abundance in fertilized soil. An ANOVA with Tukey HSD was used to determine differences in log-transformed abundance. There were no differences in the relative abundance of the bacteria with different predicted functions.

**Figure 3 microorganisms-12-01756-f003:**
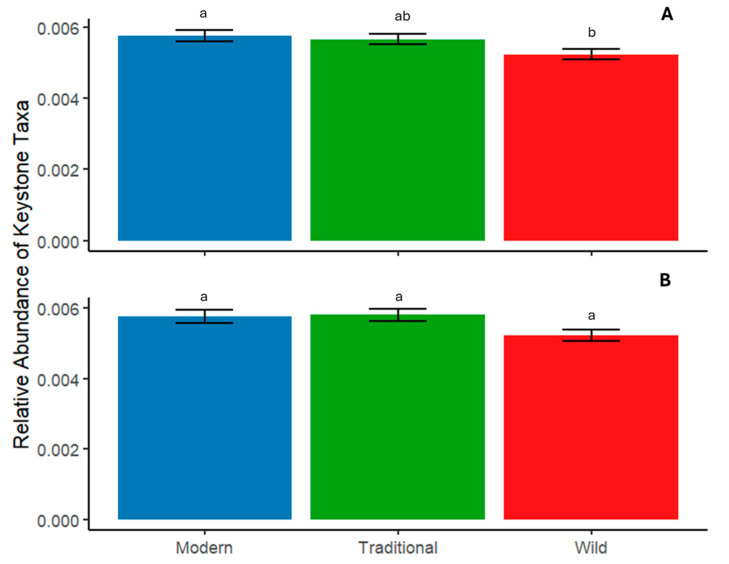
Relative abundance of the sum of keystone bacteria in the rhizosphere of wild, traditional, and modern tomato. (**A**) Unfertilized soil and (**B**) fertilized soil. Presented as mean ± SE. An ANOVA with Tukey HSD was run to determine differences. Letters denote significant differences (α = 0.05) in the domestication group.

**Figure 4 microorganisms-12-01756-f004:**
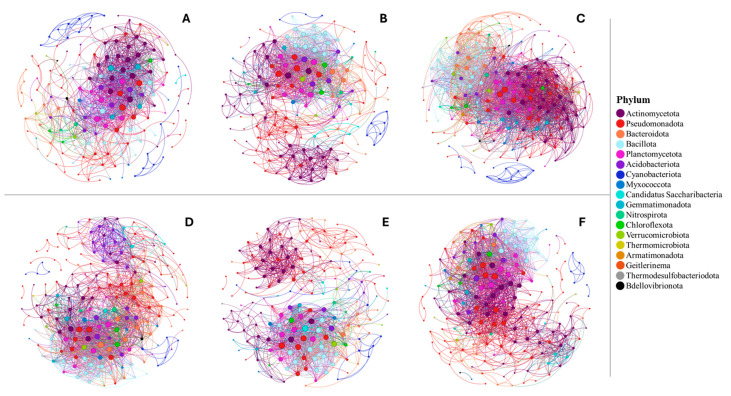
Co-occurrence network showing bacterial interrelationships in unfertilized soil of the tomato rhizosphere. Each co-occurrence network showcases correlative relationships of each tomato domestication group and fertilization treatment: unfertilized wild (**A**), unfertilized traditional (**B**), unfertilized modern (**C**), fertilized wild (**D**), fertilized traditional (**E**), and fertilized modern (**F**). Color denotes phylum. Node size is proportionate to degree (i.e., number of significant correlations). Only bacteria that were significantly correlated (optimized Spearman’s correlation > 0.7, *p* < 0.01) are included.

## Data Availability

Processed sequencing data and their corresponding codes are available at the following repository: https://github.com/marydixon/tomato_P_networks. Raw sequencing data are available in the Sequence Read Archive at NCBI: accession nos. SAMN43289197, SAMN43289198, and SAMN43289199.

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
