# Peer review of "Rhizosphere Microbiome Co-Occurrence Network Analysis across a Tomato Domestication Gradient"

_microorganisms, 2024, doi:10.3390/microorganisms12091756_

Round 1

Reviewer 1 Report

Comments and Suggestions for Authors

The manuscript, "Rhizosphere microbiome co-occurrence network analysis across a tomato domestication gradient," is exceptionally well-written and presents a very interesting set of results. The research tackles the significant question of how the plant microbiome evolves during domestication, a topic with broad implications for both agriculture and ecology. The co-occurrence network analysis employed by the authors is an innovative approach that provides valuable insights into the interactions and dependencies between different microbial populations within the rhizosphere.

There are a few minor points that could be addressed to further enhance the clarity and impact of the research.

a)       Provide a brief explanation for the authors' choice of traditional bacterial phyla nomenclature. Was this done for consistency?

b)      The plant height measurements in the methods section are unnecessary, as they are not discussed in the results or relevant to the microbiome analysis.

c)       Clarify whether all 10 replicates of each experimental condition were used for rhizosphere DNA extraction and sequencing.

d)      Although two fertilization conditions were tested, the co-occurrence network analyses were conducted only with the data from non-fertilized plants. This choice requires explicit justification.

e)      The three keystone taxa were identified from the analyses of the unfertilized dataset, and their relative abundance in the rhizosphere of fertilized plants is lower. Since key taxa identification under non-fertilized conditions was not done, the significance of the comparison is unclear. Does this imply other taxa occupy that ecological niche? Are the differences in the abundance of keystone taxa between domestication groups under unfertilized conditions statistically significant (Figure 3, dark columns)?

f)        In lines 301-307, the results are interpreted in the context of farming practices. Given that plants were grown under controlled conditions, clarify the relation between these interpretations and the results obtained.

Reviewer 2 Report

Comments and Suggestions for Authors

Review on “Rhizosphere microbiome co-occurrence network analysis across a tomato domestication gradient” for manuscript ID microorganisms-3044132

In this manuscript the authors describe the influence of domestication to rhizosphere microbiome structure. This study explores how the rhizosphere bacterial interactions in low-P conditions change with tomato domestication and breeding.

In the brief introduction authors highlight the role of soil microbes in facilitating plant adaptation to environmental stresses, such as low phosphorus availability. Additionally, the authors emphasize the impact of plant domestication on shaping the rhizosphere microbiome composition and function. Unfortunately, some recent studies have been missed. The following papers could help to improve the Intro:

·         Yu J, Wang L, Jia X, Wang Z, Yu X, et al. 2023. Different microbial assembly between cultivated and wild tomatoes under P stress. Soil Science and Environment 2:10 https://doi.org/10.48130/SSE-2023-0010

·        Namis Eltlbany et. al. Enhanced tomato plant growth in soil under reduced P supply through microbial inoculants and microbiome shifts, FEMS Microbiology Ecology, Volume 95, Issue 9, September 2019, fiz124, https://doi.org/10.1093/femsec/fiz124

Please highlight the studies dedicated to tomato microbiome changing during domestication instead of other plants, and clearly state the aim of the study.

L67: sentence about wheat could be omitted.

My questions about Results and Discussion:

Figure 1 shows the chords diagram, but only the unfertilized soil case is considered.

Figure 2. Please revise the figure layout for better clarity and use the serif font.

L292: I doubt that is correct to use Avena study, which is completely different plant, to consider relations of rhizosphere network characteristics with microbial diversity.

L146: please provide reads length and sequencing statistics.

Discussion section needs revision, while it lacks conclusion and mostly devoted to species other than tomato.

L337: ref. [62] is a review not dealing with tomato plants.

Methods section comments:

The authors obtained large amount of data, but no sequencing data were published or even mentioned to be published in future. Raw data need to be publicly available to reproduce authors’ results. The ready-to-use scenario (script) is highly affordable to add in the Github repository to produce the figures.

L177: reference [40] should be replaced with current one: https://www.nature.com/articles/s41587-020-0548-6

Some minor corrections to the text (style and spelling):

·        L354-359: please remove sample text

Round 2

Reviewer 2 Report

Comments and Suggestions for Authors

I would like to thank the authors for the improving the manuscript, but some concerns remain to be addressed.

L28-32: please avoid repetition

L51: the original work of (Prischl et al., 2012) could be cited here instead of [11]

L84: Review of studies about impact of low P on the tomato rhizobiome could be extended to better define the knowledge gap. The paper https://doi.org/10.3390/microorganisms8111844 could help along with [26] and [29].

GitHub repository is helpful, but raw sequencig data is still unavailable for the readers.
